# Can We Exploit Machine Learning to Predict Congestion over mmWave 5G Channels?

**Luis Diez** [1,*] , **Alfonso Fernández** [1], **Muhammad Khan** [2], **Yasir Zaki** [2] **and Ramón Agüero** [1]

1   Communications Engineering Department, University of Cantabria, 39005 Santander, Spain;
    alfonso.fernandezgu@alumnos.unican.es (A.F.); ramon@tlmat.unican.es (R.A.)
2   Communication Networks Lab, New York University Abu Dhabi, 129188 Abu Dhabi, UAE;
    mk7406@nyu.edu (M.K.); yz48@nyu.edu (Y.Z.)
*   Correspondence: ldiez@tlmat.unican.es

**Abstract:** It is well known that transport protocol performance is severely hindered by wireless channel impairments. We study the applicability of Machine Learning (ML) techniques to predict congestion status of 5G access networks, in particular mmWave links. We use realistic traces, using the 3GPP channel models, without being affected using legacy congestion-control solutions. We start by identifying the metrics that might be exploited from the transport layer to learn the congestion state: delay and inter-arrival time. We formally study their correlation with the perceived congestion, which we ascertain based on buffer length variation. Then, we conduct an extensive analysis of various unsupervised and supervised solutions, which are used as a benchmark. The results yield that unsupervised ML solutions can detect a large percentage of congestion situations and they could thus bring interesting possibilities when designing congestion-control solutions for next-generation transport protocols.

**Keywords:** machine learning; mmWave; 5G; congestion control; ns-3; network simulation; unsupervised learning

## 1. Introduction

Millimeter wave (mmWave) is believed to be one of the key radio technologies to cope with the capacity requirements of 5G communications [1,2]. However, due to its high frequency working range, it poses additional challenges [3]. Among others, mmWave channel exhibits high oxygen absorption, diffraction and penetration losses, which lead to highly varying physical capacity. At the transport layer, this variability might be perceived as congestion, and mechanisms could be triggered and so reduce the traffic rate. This way, transport layer protocols, such as Transmission Control Protocol (TCP), cannot fully harness the communication capacity.

Legacy loss-based congestion-control algorithms, for instance Cubic, misbehave when bottleneck buffers are large, leading to bufferbloat situations. On the other hand, they also suffer from performance hindering, when buffers are small [4]. In addition, these algorithms do not appropriately deal with varying throughput reduction over wireless links, which cause an infra-use of the corresponding resources (wireless channel).

Due to the growing interest on 5G technologies (in particular mmWave), and the performance limitations of traditional congestion-control techniques over them, we have witnessed several proposals that aim at overcoming such limitations [5–7]. Besides the potential migration issues, some of them do not yield a good performance over some 5G scenarios [8]. In particular, the authors of [9–11] have focused on the behavior of various congestion-control algorithms over mmWave.

In this paper, we take the initial steps towards the design of holistic congestion-control solutions over mmWave channels. The focus is put on how to learn the congestion status of underlying channels. This is of utter relevance, so that transport protocols might take appropriate actions based on such knowledge, which should be based on observable parameters.

It is known that mmWave channels are characterized by fading events, which might lead to long stall situations. However, once the fading situation finishes, channel capacity grows very quickly. On the other hand, traditional congestion situations are characterized by delays exhibiting rather different pace. Hence, the main goal of this paper is to explore whether observable parameters (from the transport layer) can be exploited to infer the congestion situation of underlying mmWave links. We tackle such objective by first characterizing the performance (in terms of delay) over such links, using realistic channel models in the ns-3 framework. We then analyze the relationship of such performance with the link congestion situation, by studying the corresponding correlation. Finally, we explore several machine learning techniques to establish the congestion level, using the previously identified metrics.

By tackling the aforementioned goals, the research discussed hereinafter bring the following contributions:

- We generate a set of traces that provide temporal evolution of traffic parameters, measurable at transport layer, and the congestion situation, measured as the buffer status at the access network. The traces generation process considers the complete cellular protocol stack implementation.
- We analyze the information provided by the traces to derive meaningful congestion metrics. In particular, we formally analyze potential indicators, based on their linear dependency with the congestion provided by the traces.
- Last, the most promising metrics are used to study the performance of several unsupervised machine learning algorithms that can be used to learn the perceived congestion caused by channel variations. Their performance is compared with that exhibited by supervised approaches, which are taken as a benchmark.

The paper is structured as follows. First, we discuss related works in Section 2. Then, Section 3 depicts the mmWave scenarios that we consider, and introduces the methodology we have followed to obtain the corresponding traces. In Section 4 we formally analyze such traces, and we establish metrics that can be used to learn the congestion state. Afterwards, in Section 5 we briefly describe the machine learning algorithms that we use, while we discuss their performance in Section 6. Finally, Section 7 concludes the paper, summarizing its main results, and providing an outlook of our future work.

## 2. Related Work

As was mentioned earlier, the performance of traditional loss-based congestion-control solutions is severely hindered by channel variations that characterizes wireless communications. To overcome this limitation, new delay-based algorithms have recently loomed. For instance, the authors of [5] proposed Verus, which is built upon learning the relationship between the perceived delay and transmission window in cellular networks. Similarly, Sprout [6] proposed to infer wireless channel variation in cellular networks from packet inter-arrival times, and to adapt the transmission rate accordingly. Following a different approach, Performance-oriented Congestion Control (PCC) Vivace [7] exploits online convex optimization to optimize the sender rate. Although all these schemes have been shown to offer good performance over cellular networks, their behavior worsens [8] over certain highly variable 5G scenarios, like mmWave.

On the other hand, there are other works that have analyzed the performance of congestion-control solutions over mmWave channels, either through simulations or using real platforms. For instance, Zhang et al. analyzed in [9] the performance of different TCP algorithms exploiting a ns-3 mmWave module. This study evinced a bad adaptation of the congestion-control solutions, which led to frequent slow-start phases and bufferbloat situations. This work was further extended

in [10], were more congestion-control algorithms were analyzed, including Bottleneck Bandwidth and Round-trip propagation time (BBR). This latter paper discusses the tight interplay between transport layer performance and the particular network configuration (for instance, packet size, 3GPP stack parameters, etc.). Other works have tried to analytically model this relationship. For instance, authors in [11] analyze the performance of TCP-Cubic over 5G mmWave channels, using finite-state Markov Chains to model the system behavior. To complement the results obtained by simulators and analytical models, the performance of congestion-control solutions over mmWave has been also studied using real hardware. In this sense, the measurement campaign depicted in [12] also revealed a bad adaptation of both Cubic and BBR to mmWave channel variations. The performance of BBR was also analyzed in [13], which shows a severe throughput degradation over mmWave channels. The authors even reported what they referred to as "throughput collapse", for bad Round-Trip Time (RTT) estimations.

The authors of [14] thoroughly analyze bufferbloat in 5G mmWave scenarios and they point out that existing congestion-control solutions are not able to adapt to rapid rate variations. In addition, their work shows that Active Queue Management (AQM) techniques can mitigate this phenomenon. Altogether, all previously mentioned papers foster the idea of developing adaptive solutions, able to learn the congestion level, for instance exploiting machine learning techniques.

In this sense, the adoption of machine learning-based solutions, in the context of 5G networks, is becoming quite popular, and the scientific community is fostering the development of data-driven algorithms, which may complement traditional model driven approaches [15,16]. More importantly, artificial intelligence in general, and machine learning in particular, is believed to be one of the key enablers for future 6G networks [17–19].

Another category of works aim to propose new transport layer solutions (clean state approach) or to adapt existing congestion-control algorithms to improve their performance over mmWave networks. For instance, Na et al. propose in [20] a deep-learning-based solution for TCP congestion control in disaster scenarios with high-mobility drones. Although they share some of our goals, they are focused on a very particular scenario and its solution, while our main goal is to ascertain the potential performance of different machine learning algorithms. Similarly, the impact of mobility over TCP in mmWave scenarios has been addressed in [21]. The authors point out the importance of efficiently managing mobility to improve the TCP performance in terms of both throughput and delay. In addition, they evince that even with a very efficient mobility management scheme, TCP is very sensitive to end-to-end delay, which strongly impacts both users and network.

Although 5G mmWave would certainly boost the capacity of forthcoming cellular network deployments, it will coexist with other technologies, such as LTE. In these heterogeneous scenarios traffic could be offloaded to LTE networks, if mmWave coverage degrades. This possibility has been studied in different works by exploiting multi-path TCP (MPTCP). For instance, Lee et al. [22] propose an offloading technique (offloading by restriction - OBR) to optimize the throughput in 5G and LTE heterogeneous networks. Following a similar approach, the performance of Linux MPTCP over LTE and mmWave links is analyzed in [23]. In particular, the authors seek the optimal configuration of secondary paths and how to select the corresponding congestion-control algorithms. Furthermore, Sahe et al. analyze in [24] different coupling solutions for MPTCP over dual-band 60GHz/5GHz links, using off-the-shelf hardware. As can be seen, the goal of these works is different to ours, since we do not consider multi-path. On the other hand, none of them exploit machine learning techniques. We will consider the integration of multi-path within the proposed learning scheme in our future work.

Although TCP, and its various modifications and extensions, stands out as the most widespread solution at the transport layer, Quick UDP Internet Connections (QUIC) has recently appeared as an appealing alternative, especially for web-based traffic. However, the negative impact of wireless channels over congestion-control algorithms persists. In this sense, Sinha et al. introduced in [25] a cross-layer QUIC enhancement to optimize handover decisions considering wireless channel variations in 5G mmWave scenarios.



It is also worth noting that there exist a few works that have applied machine learning both to predict the congestion level and to develop novel congestion-control solutions. For instance, Geurts et al. analyzed in [26] the performance of different machine learning algorithms to build a loss classifier over random topologies. In this regard, their research shares some of our goals. However, it was not applied to mmWave networks, while we focus on delay, rather than on loss.

In addition, the authors of [27] analyze congestion learnability in general, but they do not consider the particular features of mmWave channels. On the other hand, Ahmad et al. [8] addresses congestion learnability over mmWave networks, but the network emulation that was used does not consider the relationship between congestion and other protocol stack elements from the cellular access network.

All in all, we can conclude that the use of machine learning techniques to learn congestion situations over mmWave channels has not been thoroughly addressed yet.

## 3. Scenarios and Dataset Generation

In this section, we describe the process followed to obtain transport layer traces from communications over mmWave channels. In the literature we can find different approaches to obtain such traces. For instance, the authors in [28] create a 5G trace dataset from a major Irish operator. This dataset can be afterwards used to feed tools, such as Mahimahi [29], to emulate end-to-end communication. However, this approach has two main limitations. First, the obtained traces could have been impacted by the particular congestion-control algorithm used by the transport layer protocol (quite likely TCP-Cubic). Hence, the metrics that could be extracted from such traces would not accurately reflect the most appropriate raw input to design a congestion-control algorithm. Besides that, as was mentioned before, existing studies point out the strong impact of the cellular protocol stack configuration over the congestion-control solution behavior. In this sense, real traces would hide such configuration, thus allowing less control over the analyzed scenario.

As an alternative, we obtain the traces exploiting the ns-3 network simulator [30], and in particular the mmWave module [31], which has been used in the past to carry out end-to-end simulations [32,33]. This module provides mmWave channel models, including those defined by the 3GPP [34], and novel ones proposed in the literature [35]. In addition, the module is equipped with a complex channel model [36], which allows performing realistic simulations. On the other hand, it has been extended to allow dual connectivity [37] and carrier aggregation [38], which would allow us to analyze how the simultaneous use of different wireless channels might impact the performance of congestion-control solutions. In this work we focus on the analysis of different machine learning algorithms (and their potential) over a single mmWave channel, leaving multi-channel scenarios as an extension that we will tackle in our future work.

In the following we describe the main characteristics of the analyzed scenarios, and we discuss the obtained parameters, which will be used in the following section to derive congestion metrics. In addition, we also study the impact that network scheduling policy may have over the transport layer performance.

### 3.1. Scenario Setup

The evaluation scenario comprises one user connected to a mmWave base station, as depicted in Figure 1. Although having a greater number of users in the scenario may eventually impact the end-to-end performance, our interest lies on assessing whether it is feasible to learn the perceived congestion due to mmWave channel variations. In this sense, having a single user allows us to isolate the effects that the wireless channel variability would have over user traffic, especially if there were more active communications. More complex scenarios will be considered in the future, once the impact of the channel has been understood and characterized.

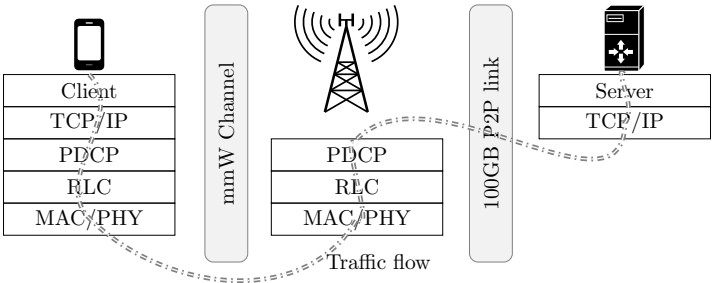

**Figure 1.** General description of the scenario under analysis.

Although there exist different channel models [39,40], in this work we focus on the ones defined by the 3GPP [41]. In particular, we apply the urban macro channel model, where both line-of-sight and non-line-of-sight situations may happen, according to the distance between user and base station [41]. In our scenario the user remains static, at a given distance from the base station, while the corresponding channel matrix is updated every millisecond. For the sake of simplicity, we assume that the antenna beam in the transmitter is perfectly aligned towards the receiver.

As can be observed in Figure 1, we do not consider any limitation in the network from the base station to the application server, which is implemented using a 100 Gbps point-to-point link, but only in the cellular access network. To model the congestion at the wireless segment, caused by mmWave channel variations, we monitor the evolution of the buffers in the protocol stack. In particular, we focus on the Packet Data Convergence Protocol(RLC) transmission buffer, which increases when the number of transmission opportunities to the physical layer decreases, or when the Modulation and Coding Scheme (MCS) needs to be adapted due to poor channel conditions.

Due to the relevance and potential impact of the RLC protocol in our study, it is worth discussing its purpose and possible configurations. Among other tasks, it is responsible for segmentation and reassembly of RLC Service Data Units (SDUs) coming from the Packet Data Convergence Protocol (PDCP) layer, see Figure 1. In addition, it ensures duplication detection and deletion, as well as in-order frame delivery to the upper layer. When configured in the Acknowledged Mode (AM), RLC implements Automatic Repeat Request (ARQ), to provide reliable link-level communications, so that the connection finishes if the maximum number of retransmissions is surpassed. As for the MAC and link layers, we enable Hybrid Automatic Repeat Request (HARQ) in our setup, and RLC protocol is configured in the AM mode, since it is the most common choice. It is worth noting that with this configuration, losses over the wireless access segment are very unlikely. Hence, congestion-control mechanisms would be mainly triggered due to delay (i.e., Retransmission Timeout-RTO timer expiration in TCP), provided sufficiently long buffers exist in the rest of the network, thus ensuring there are no losses.

At the application layer (user traffic) we consider a constant bit rate pattern, using User Datagram Protocol (UDP) as transport protocol, to avoid having congestion-control actions from happening, since they would impact the trace evolution, as was mentioned in Section 2. In this sense, the reason to use UDP is that our interest is the analysis of the relationship between congestion on the wireless access segment with the metrics that may be observed at the transport layer. Indeed, the use of UDP allows us to obtain raw information to study the learning possibilities of the different machine learning algorithms. If, on the other hand, we had used TCP, the particular congestion-control algorithm applied would have affected the analyzed parameters and the buffer status. For instance, slow-start phases due to bad channel conditions would lead to remarkable decrease of the traffic volume at the wireless access network. In turn, the RLC buffer would empty indicating that there is not congestion in the access network. On the other hand, UDP traffic sending is not affected by channel fluctuations, so that it would continue transmitting data leading to a congestion event.

We have obtained traces for different distances and data rates, and for each configuration we record parameters from different layers, as follows:

- Physical layer: although the mmWave physical channel is actually seen as a black-box to the upper layers, we measure its evolution, to understand how its capacity variation translates to upper layers metrics. In particular, we monitor the amount of data sent to the physical channel as the transport-block-size that is actually used in each transmission opportunity. This parameter is relevant to understand how some network configurations, such as the scheduler policy, may impact the overall system behavior, including congestion-related metrics at the transport layer.
- MAC layer: in this layer we specifically focus on the RLC protocol. Congestion at the access segment caused by physical channel fluctuations translates into a higher buffer occupancy at the access-stratum protocol stack. Although other buffers exist at the physical layer to implement the HARQ, it is the RLC level where most of the buffering takes place. We monitor the evolution of the RLC buffer length, as an indicator of the level of congestion at the access network. We estimate it as the difference between bytes received at the protocol instance and those that were sent and acknowledged, so that it considers the actual occupancy of transmission and retransmission buffers. In our scenario configuration, RLC buffers are considered to have an infinite capacity, so that RLC SDUs are never dropped.
- Transport layer: considering that losses are rather unlikely, congestion is reflected at the transport layer by a delay variation. For that reason, we registered the packet delay as well as the Inter-arrival Time (IaT). In particular, the packet delay is measured as the time between the moment the packet left the application sender until it is successfully received at the destination. On the other hand, the IaT is measured as the time between two consecutive packet receptions. Although these parameters would not be directly accessible from the congestion-control algorithm, both can be easily estimated by means of the Round-Trip Time (RTT) and the time between ACK segments, respectively.

*3.2. Scheduling Policy Impact*

The scheduler in cellular networks is responsible for adapting the MCS to the channel conditions. Furthermore, in Orthogonal Frequency-Division Multiple Access (OFDMA)-based cellular networks the scheduler also assigns physical resources to traffic flows. Therefore, the scheduling policy may have a strong impact on the actual transmission rate and on the RLC buffer variation, which is our congestion indicator. In the following we depict the evolution of the parameters that we monitor, to analyze how they are affected by the scheduling policy. In particular, we have used two different schedulers. The first one is a greedy scheduler that allocates radio resources, according to the RLC buffer status and the corresponding MCS, to maximize the throughput, while the second one is the well-known max-weight algorithm [42].

Figures 2 and 3 show the instantaneous evolution of the different parameters obtained using both schedulers for the very same wireless channel instance. The results have been obtained during 3 s, in a scenario where the user is 30 m away from the base station, and the traffic rate is set to 50 Mbps. We represent the amount of data sent to the physical channel and the RLC buffers' occupancy in kB, as well as the packet delay and IaT in milliseconds.

As can be observed in Figure 2, the greedy scheduler sends the traffic to the physical channel in bursts, leveraging those periods with good channel conditions (i.e., line-of-sight). This yields a clear variation of the RLC buffer occupancy, which increases when the transmitted traffic is low, and gets empty periodically, concurring with longer physical transmissions. In turn, the results also evince that the delay observed at the transport layer follows a similar pattern to that of the RLC buffer. Opposed to that, the IaT exhibits a less predictable behavior. The relationship of the IaT with the congestion level will be further analyzed in Section 4.

Then, Figure 3 shows that the max-weight scheduler yields a different behavior. Again, the traffic is sent in bursts, but with different periodicity and burst duration. As can be seen, it leads to different patterns of the RLC buffer length, as well as of the parameters measured at the transport layer. Nevertheless, in both cases the delay seems to follow the buffer growing trend while the IaT presents a different behavior.

Despite the differences seen for both schedulers, the relationship between RLC buffer and delay-related metrics is clear for both of them, and it is the pace at which the RLC buffer changes where such difference stands out more clearly.

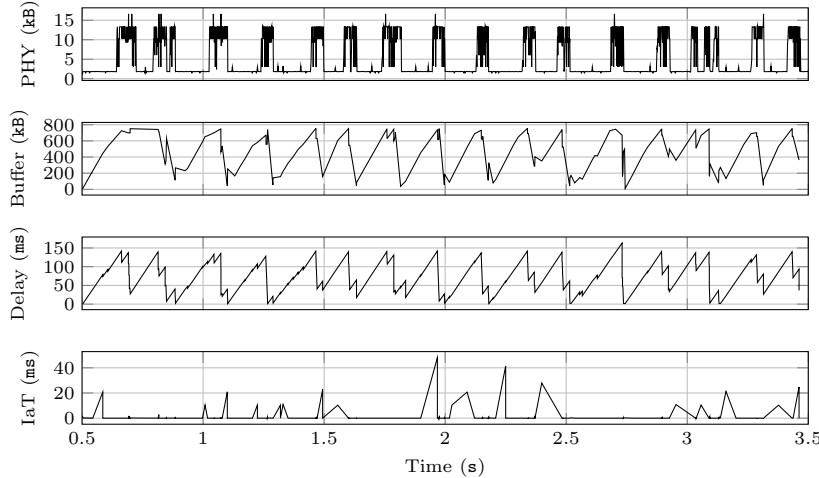

**Figure 2.** System parameters with distance and traffic rate of 30 m and 50 Mbps, respectively, using the greedy scheduler.

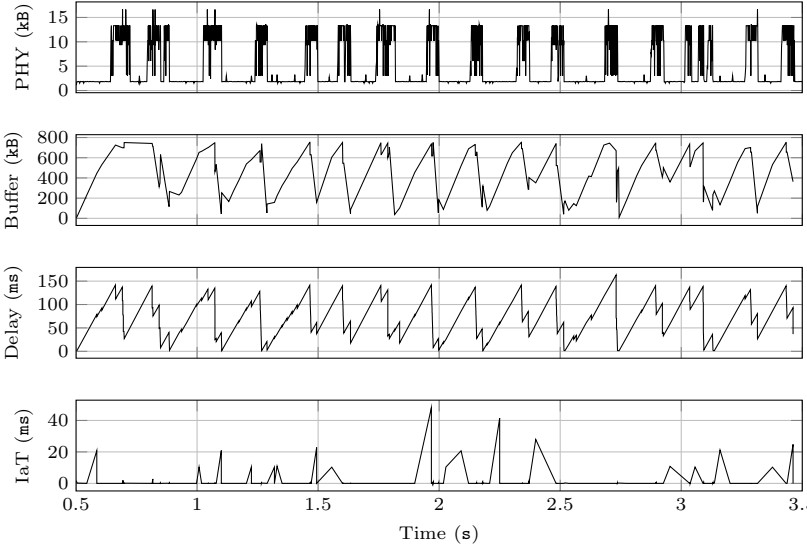

**Figure 3.** System parameters with distance and traffic rate of 30 m and 50 Mbps, respectively, using max weight scheduler.

## 4. Dataset Analysis

After describing the scenario setup, we herewith study the relationship of both transport layer parameters (i.e., delay and IaT) with the congestion level (RLC buffer size). This analysis allows us to develop meaningful metrics to learn the congestion status. For that we first study the linear correlation of the raw observed parameters. Then, based on the corresponding results, we identify and analyze various congestion metrics. Hereinafter, we will use correlation to refer to the Pearson's correlation coefficient, which is defined, for two random variables $X$ and $Y$, as $\rho_{XY} = \frac{cov_{XY}}{\sigma_X \sigma_Y}$, where $cov_{XY}$ is the corresponding covariance, and $\sigma_X$ and $\sigma_Y$ represent the standard deviations of $X$ and $Y$, respectively. Although the congestion value would depend on the particular implementation of the RLC buffer, in this work we have assumed that there exists congestion when the buffer exceeds 0.5 MB. This value corresponds, approximately, to 340 packets of 1472 `bytes` waiting, at the user equipment, to be transmitted.

### 4.1. Correlation Analysis

Figures 4 and 5 illustrate the linear correlation of the RLC buffer occupancy with both the delay and the IaT for the greedy and max-weight schedulers, respectively. These results have been obtained over scenarios with different traffic rates, and distances between user and base station (as shown in the x-axis), for connections lasting 30 s. Traffic rates are chosen in the range $[10, 1000]$ Mbps, and the distances are set to ensure that communication was not dropped due to timer expiration.

Figure 4 shows the results that were obtained with the greedy scheduler. First, we can observe in Figure 4a that for data rates below 50 `Mbps`, the correlation between delay and congestion level is rather high for distances above 20 m. In addition, we can also observe that the correlation drops as we increase the application data rate. The results show that for data rates above 100 `Mbps`, the correlation stays below 0.5.

Worthy of explanation is the behavior yielded by scenarios with short distances (10 and 20 m). In these cases, the probability of having line-of-sight conditions is very high, and thus the channel presents a large and stable communication capacity. Under these circumstances the buffer variation (albeit rather slight) is not related to capacity fluctuations, and the correlation values do not provide meaningful information. As can be observed in Figure 4, in general the correlation for short distances is very low, but for scenarios with 30 `Mbps` we can observe larger values. Similar behavior can be observed when using the max-weight scheduler, in Figure 5, for scenarios with 40 `Mbps`. All in all, correlation values for such short distances are not useful to predict the congestion state.

Then, Figure 4b depicts the correlation values observed for the IaT. In this case, the results evince that the raw value of the IaT does not actually follow the buffer growing trend, as was already anticipated by the results discussed in Section 3. Regardless the distance or traffic rate, the values obtained are below 0.1, but for just one configuration.

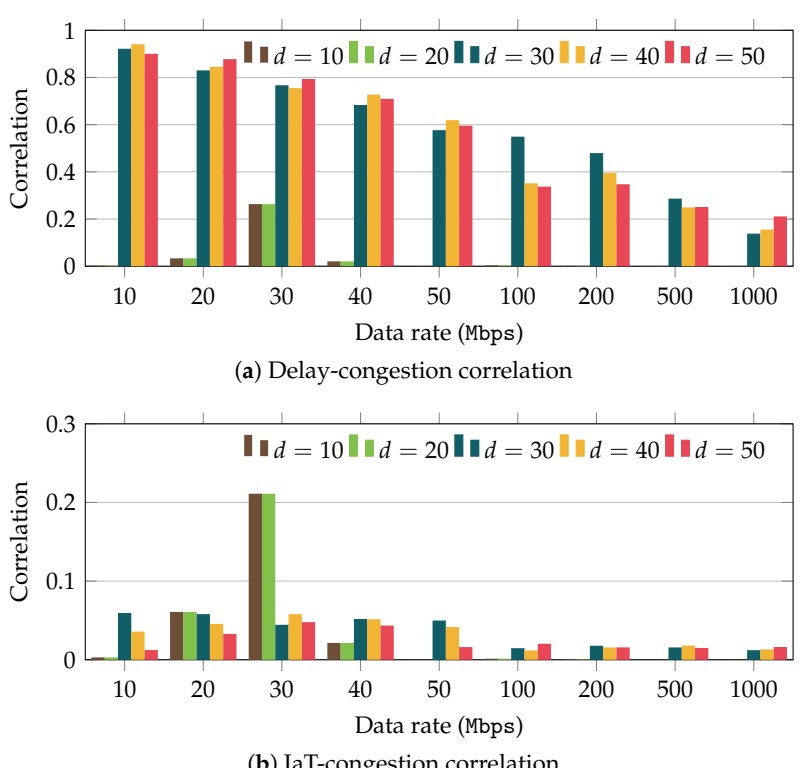

**Figure 4.** Linear correlation between the RLC buffer and transport level metrics using the greedy scheduler.

Then, Figure 5 shows the results that were obtained when we used the max-weight scheduler. As can be observed in Figure 5a the delay correlation exhibits a different trend, when compared with the one seen for the greedy scheduler. Although for low and medium data rates the correlation

is significantly lower, as we increase the traffic rate, it grows remarkably, even for small distances between user and base station. On the other hand, if we analyze the correlation corresponding to the IaT, Figure 5b, we can see a trend rather similar to that observed with the previous scheduler.

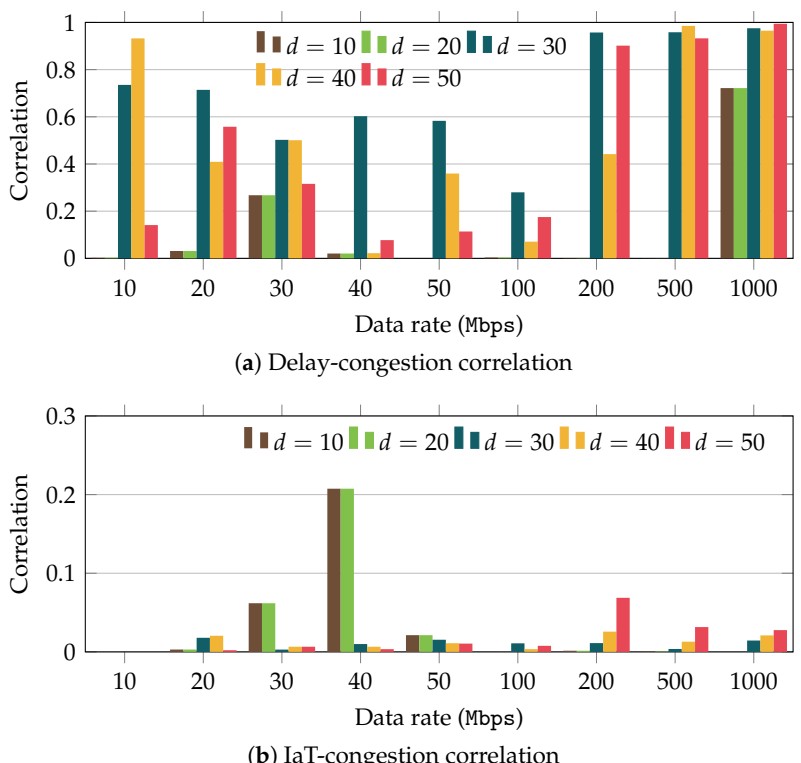

(**a**) Delay-congestion correlation

(**b**) IaT-congestion correlation

**Figure 5.** Linear correlation between the RLC buffer and transport level metrics using the max-weight scheduler.

These results indicate that as could have been expected, there exists a clear relationship between the congestion at the access segment and the delay experienced at the transport layer. Nevertheless, such relationship strongly depends on both the distance and on the application data rate. Furthermore, we have also observed that the usage of the physical resources established by the scheduler has a remarkable impact on such correlation. On the other hand, the raw IaT value does not yield a clear correlation with the congestion. In the following we will exploit the raw values of both IaT and delay to build derived metrics. Although the particular scheduler has some impact, we will restrict the results for the greedy scheduler for simplicity.

### 4.2. Metrics Definition and Analysis

Besides the raw parameters obtained from the scenario traces (delay and IaT) we have also identified a set of simple metrics, derived from such parameters, which may be used to improve the behavior of congestion learning. Apart from their accuracy, it becomes important to keep these metrics simple, so they could be computed by less capable devices. For that reason, we have focused on moving averages and standard deviations. It is worth noting that similar metrics have been already used to implement congestion-control solutions. For instance the soft-RTT calculation in [43] is estimated as a cumulative moving average, while the average standard deviation is used in the well-known Jacobson algorithm [44], as an approximation for the actual standard deviation in the RTT estimation.

In particular, the following metrics have been studied:

- $\delta$: raw value of the delay.
- $\iota$: raw value of the IaT.
- $\bar{\delta}$: Moving average of the delay.

- $\bar{\iota}$: Moving average of the IaT.
- $\sigma_\delta$: Moving standard deviation of the delay.
- $\sigma_\iota$: Moving standard deviation of the IaT.

For the moving average metrics, we exploit the Simple Moving Average (SMA) method. It is defined as the unweighted mean of $\omega$ samples, where $\omega$ is the sample window. For a sequence of samples of a variable $X$, we define the moving average obtained at the $n^{th}$ sample as $\overline{X}_n = \frac{1}{\omega} \sum_{i=0}^{\omega-1} X_{n-i}$. Similarly, the moving standard deviation obtained for the $n^{th}$ sample of a variable $X$ is defined as $\sigma_{X_n} = \sqrt{\frac{\sum_{i=n-\omega+1}^{n} \left(X_n - \overline{X}\right)^2}{\omega-1}}$, where $\omega$ is again the corresponding window length.

In Figure 6 we illustrate the correlation values obtained for the different metrics when we fix either the distance or the data rate. In particular, Figure 6a shows the results observed for a distance of 40 m and different data rates, and Figure 6b depicts the performance obtained with a data rate of 50 Mbps and for various distances. The former results show that the delay correlation is much higher than that yielded by the IaT. However, the average value of the IaT ($\bar{\iota}$) seems to be a better metric than the one corresponding to the delay ($\overline{\delta}$). Then, if we look at the moving standard deviation, we can see that for both delay and IaT there is a clear correlation with the congestion, especially for low data rates. In the case of the delay, this behavior was expected, as the delay metric is strongly related to the congestion; however, that trend was not observed for the IaT. On the other hand, we saw in Section 3 that the IaT exhibited some peaks, in many cases aligned with times when the buffer started to grow. This behavior is the reason of the high correlation values between the moving standard deviation of the IaT and the congestion experienced in the buffer, which at the transport layer is perceived as *jitter*.

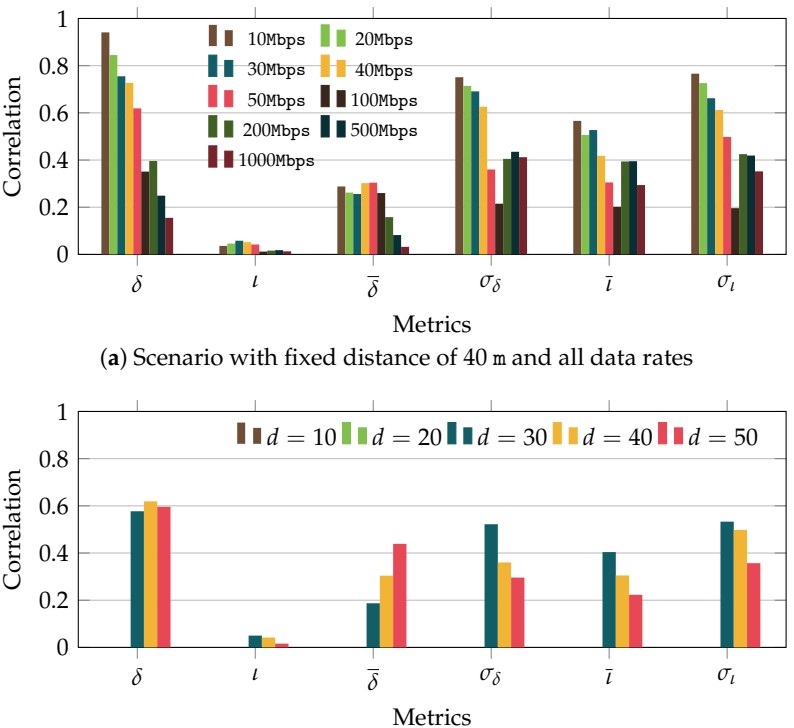

(**a**) Scenario with fixed distance of 40 m and all data rates

(**b**) Scenario with fixed data rate of 50Mbps and all distances

**Figure 6.** Correlation of all metrics with congestion.

The scenarios with fixed data rate, shown in Figure 6b, yield a similar trend. The moving average metric improves the correlation of the IaT, while it is not a good option for the delay. In addition, the moving standard deviation of the IaT shows rather good results, as compared with its raw value . It is worth noting that as we observed in the previous section, the correlation values for short distances

(10 and 20 m) are very low, since the user experiences very good channel conditions, so small buffer variations do not impact transport layer performance.

Finally, we have also analyzed the cross-correlation between the different metrics, to know whether their combined use would yield additional benefits to the learning process. In Figure 7 we plot the cross-correlation matrix of all metrics (including the congestion itself) for a distance and data rate of 30 m and 30 Mbps respectively. It is worth noting that the same relationship has been observed for other scenarios. The figure shows a matrix where each cell indicates the correlation between the metrics in the corresponding row and column. As mentioned before, we include the congestion metric itself (with symbol $\zeta$), so the figure yields both the correlation with the buffer congestion, as well as among the different metrics. For instance, the correlation between the congestion and average delay ($\zeta$ and $\delta$ respectively) corresponds to cell (1,2). Since the matrix is symmetric, the same result can be seen in cell (2,1). To make the results more visual, we use different colors and circle sizes the correlation values of each pair of metrics, being the diagonal the results corresponding to the auto-correlation of each metric (which equal 1). As can be seen, the greatest correlation value is obtained between delay and congestion. Furthermore, the moving standard deviations of both delay and IaT also show high values. Nevertheless, if we observe the relationship of the delay metric ($\delta$) and those related with the moving standard deviation, we can also perceive high correlation (row 2 with columns 5 and 7). According to these observations, we can conclude that the use of the moving standard deviation metrics would not add much information to the learning process.

Based on this analysis, the learning evaluation will be based on: (1) delay metric $\delta$, and (2) moving standard deviation of the IaT. Including this latter metric would in addition allow us to consider *jitter* information.

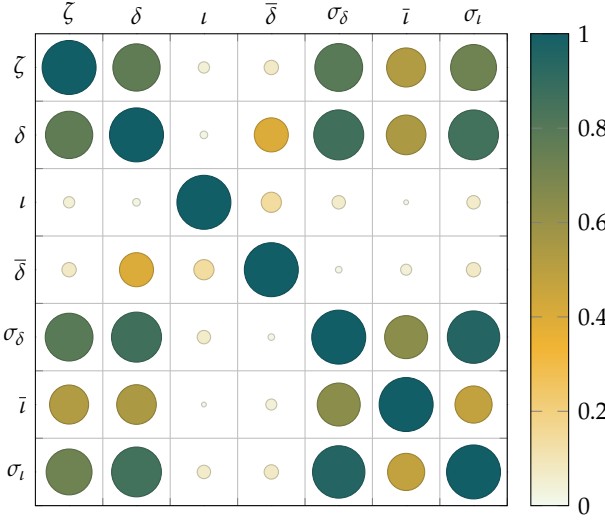

**Figure 7.** Summary of correlation of all metrics with congestion for all the scenarios analyzed, for fixed distance and data rates of 30 m and 30 Mbps. $\zeta$ is used for the congestion metric itself.

## 5. Methods for Congestion Learning

In this section, we discuss the algorithms that have been evaluated for congestion prediction, exploiting the metrics that were identified in Section 4. The methods that have been considered include both supervised and unsupervised learning techniques. Although the main goal is the development of an unsupervised approach, the supervised solutions will be mainly evaluated for comparison purposes (they are appropriate choices for our benchmark). It is worth mentioning that instead of targeting a continuous congestion prediction, we aim to classify the congestion according to several discrete levels. In particular, we have addressed binary classification, and multi-level congestion classification. The analyzed methods take as input samples of the defined metrics and group them in clusters that correspond to a predicted congestion level. In the case of unsupervised learning, once the classification

is obtained, the labels are assigned based on the average delay level of each cluster, so that longer delay values would correspond to higher congestion.

For the unsupervised learning algorithms we have mainly focused on those with low complexity for a two-fold reason. First, practical ML solutions for congestion-control algorithms should be simple, since it takes place in short temporal intervals and over devices with varying capabilities. In addition, after confirming that there exists a relationship between congestion at the access network and transport layer parameters, our objective is to assess the feasibility of applying ML solutions to improve congestion-control techniques. The use of more complex or specific ML algorithms and configurations are left for our future work.

### 5.1. K-Means

It is one of the most widespread clustering algorithms, due to its simplicity. The algorithm aims to minimize the Within-Cluster Sum of Squares (WCSS). Each cluster is defined by the average value of its samples, which is referred to as centroid. Then, the algorithm seeks to minimize the summation of the squares of distances from each point to its corresponding centroid, as indicated in Equation (1):

$$argmin(S) \sum_{i=1}^{k} \sum_{x \in S_i} ||x - \mu_i||_2^2 \tag{1}$$

where $k$ the number of clusters, $S$ is the set of samples, $S_i$ are those samples assigned to the $i^{th}$ cluster, and $\mu_i$ the current centroid of such cluster. $|| \cdot ||_2$ corresponds to the Euclidean norm. This algorithm has some limitations when the optimal clusters cannot be easily separated. On the other hand, its low computational complexity would simplify its implementation in any device.

### 5.2. Hierarchical Clustering

Hierarchical clustering is considered when we assume a multi-level congestion prediction. There exist different ways of doing hierarchical clustering. We can start from a single cluster and divide it further, following a top-down approach, or we can follow a bottom-up process, starting from a high number of clusters that are combined. In this work we have followed the former alternative, to know at which extent we can split the samples in clusters, so that they actually represent congestion levels.

To select the number of clusters, the different sets of data have been analyzed. As an example, Figure 8 shows the dendrogram of the dataset for the scenario with data rate of 50 Mbps and distance of 50 m; similar results were obtained for other configurations. In this plot, the similarity of two clusters is measured by the height where they join, so that the lower the clusters join, the more similar they are. As can be seen, many clusters are quite similar (they join at low levels), which would prevent algorithms from having a good performance. Considering these results, we decided to configure the classification with multiple clusters to 4 congestion levels. For the implementation of this algorithm we have used *k-means* in an iterative way, so that it performs binary classification over the obtained clusters.

### 5.3. Hidden Markov Model (HMM)

It is a statistical model widely used for temporal series. There exists a set of internal states that cannot be directly observed, while they generate several observable variables. It is assumed that the internal states follow a first order Markov Chain. This algorithm has two main applications:

- If the model is known, it is possible to ascertain the most likely state sequence that led to a given observable sequence.
- Given a sequence of observable samples, it is possible to estimate the transitions between states, as well as the model parameters. This problem is typically solved using the Baum-Welch algorithm, also known as Expectation-Maximization (EM).

Congestion learning corresponds to the second use case, where given a sequence of metrics, we aim to estimate the state transitions (in terms of congestion) that caused it. The reader may refer to [45] for a succinct discussion of HMMs and their possible implementations.

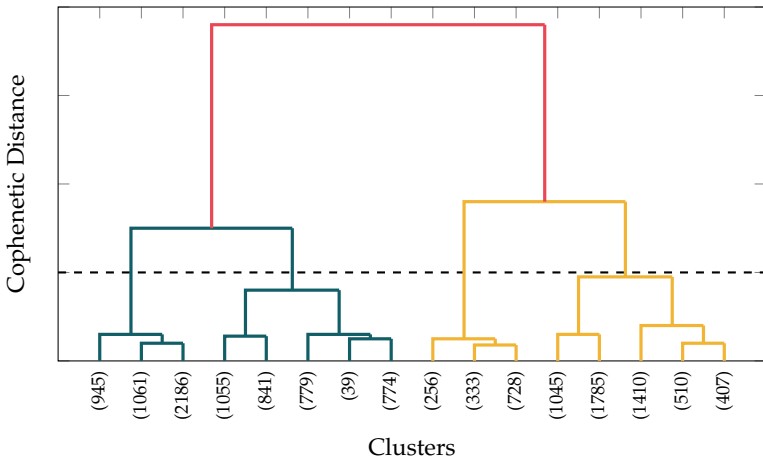

**Figure 8.** Hierarchical clustering dendrogram for a scenario with a data rate of 50 `Mbps` and 50 m.

### 5.4. Support Vector Machine (SVM)

It is a family of supervised algorithms applied to a broad range of tasks, such as regression or classification, where it is usually called Support Vector Classification (SVC). In the case of binary classification, SVC defines a hyperplane, so that the distance to the closest point of each cluster (support-vectors) to such hyperplane is maximized. This methodology can be extended to multi-level classification following the strategy "one-against-one", where various classifiers are trained for each pair of levels. This way, for $n$ clusters, we would need $n \times \frac{n-1}{2}$ classifiers, each trained in a binary fashion. This algorithm has been used for comparison purposes to know how far the unsupervised learning solutions performances are from the supervised version (benchmark). In all cases, the SVC implementations have been trained with the 15% of the samples, while the remaining 85% have been used for evaluation.

### 5.5. Recurrent Neural Networks (RNN)

Considering the growing relevance of neural-network algorithms, we have also evaluated the performance of this type of solutions. In particular, we have exploited a Recurrent Neural Network (RNN), with Long Short-Term Memory (LSTM) neurons. The model consists of two LSTM layers, of 100 neurons each, followed by a *fully connected* layer embracing 50 neurons. Opposed to previous algorithms, this solution yields an actual congestion value, rather than a classification. The predicted continuous values are then grouped, so that the performance of the RNN can be compared with those obtained by the other solutions.

## 6. Learning Analysis

In this section, we present the main results obtained by using the Machine Learning algorithms identified in Section 5 to predict the congestion level over the different scenarios. The evaluation has been done for both binary and multi-level congestion classification, and unsupervised approaches have been compared with the performance yielded by the supervised solution based on SVC, which is thus used as a benchmark. In particular, for binary classification we have compared *k-means* and SVC, while in the multi-level case SVC is compared with the hierarchical solution, HMM and RNN. Before running the different algorithms, the dataset was scaled, ensuring that both metrics, delay and moving IaT standard deviation, are comparable. Each feature was scaled to be in the $[0, 1]$ range. Although different weights can be given to each parameter, in this work both features are given the same relative weight.

### 6.1. Binary Classification

For the binary classification we compare the performance of *k-means* and SVC, based on the following parameters:

- Precision: it is the ratio of correctly predicted positive observations to the overall positive observations. It shows the probability that a predicted congestion event corresponds to an actual one, so it yields how trustworthy the positive prediction is.
- Recall (sensitivity): it is defined as the ratio of correctly predicted positive observations to all actual positive observations. This metric tells us the probability of detection of a congestion event.
- F1-score: it combines the two metrics above. It is typically used when both metrics, precision and recall have comparable weights, and it is defined as $F_1 = 2 \cdot \frac{\text{Prec} \times \text{Recall}}{\text{Prec} + \text{Recall}}$

In Figure 9 we represent the performance of both *k-means* and SVC for a distance of 40 m between user and base station, and for different data rates. The *k-means* performance in Figure 9a yields interesting results. First, for low data rates the unsupervised algorithm provides a reasonably good performance. The precision is high, so that the probability of having false positives is rather low. In addition, we can observe that such precision decreases as we increase the data rate, which matches the conclusions of the correlation analysis discussed before. In addition, the results indicate that most of the congestion events are successfully detected, as can be inferred from the recall values. As can be seen, the recall metric shows a similar trend as that observed for the precision, so its value decreases for higher values of the application data rate.

Figure 9b compares *k-means* with the supervised learning approach, SVC. We can observe that both solutions yield similar performance for low rates. However, when the traffic rate is high, SVC clearly outperforms the unsupervised learning, as could have been expected. Although in some cases, we could have inferred that *k-means* performs better than SVC under some circumstances, the *F1-score* of the latter is always higher.

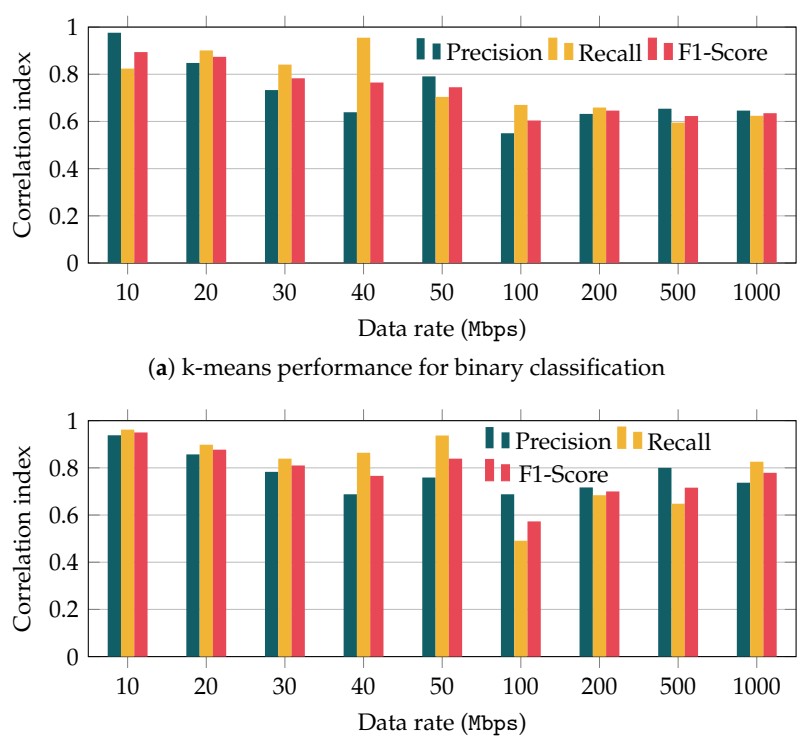

(**a**) k-means performance for binary classification

(**b**) SVC performance for binary classification

**Figure 9.** Performance of binary classification for a distance of 40 m and the different data rates.

### 6.2. Multi-Level Classification

As mentioned before, the multi-level classification for the different algorithms has been configured for 4 congestion states: the higher the state, the higher the congestion. To simplify the comparison of the different solutions, we have defined a pseudo-precision metric that penalizes how far the predicted level is from the real one. The reasoning behind this metric is establishing a practical means to measure the error of the predicted value. For instance, in a situation where the actual congestion state is 1 (low congestion), predicting a state 4 (highest congestion) is considered to be a worse result than predicting state 2. This pseudo-precision parameter is defined in Equation (2):

$$\text{pseudo-Prec} = \sum_{i \in \mathcal{S}} \frac{1}{S_i} \left( P_i + \sum_{j \in \mathcal{S}/i} \frac{P_j}{2^{|i-j|}} \right) \tag{2}$$

where $\mathcal{S}$ is the set of congestion levels. Then, $S_i$ and $P_i$ are the actual and predicted set of samples for state $i$, respectively. This way, for each congestion level $j$ different to the actual one, the contribution of the predicted samples to the precision decreases exponentially as the predicted level is further from the real one. For instance, defining 4 levels the contribution of level 2 to the pseudo-precision metric is defined in Equation (3)

$$\text{pseudo-Prec}_2 = \frac{P_2 + 0.5 \cdot (P_1 + P_3) + 0.25 \cdot P_4}{S_2} \tag{3}$$

so that when all the predicted values are correct (i.e., $S_2 = P_2$), the precision reaches 1. Although more accurate metrics could be used, this one allows us to compare the performance of the different multi-level congestion learning approaches.

First, Figure 10 shows the confusion matrix of the results obtained with the hierarchical clustering. It illustrates the behavior of the algorithm that would be otherwise overlooked using the pseudo-precision metric. As can be observed, while the congestion detection probability, which corresponds to the diagonal elements, is rather high, the prediction concentrates in the closed-by states. For instance, the first congestion level $l_1$ is correctly predicted with probability 0.6. However, wrong predictions concentrate in the second level ($l_2$) and, with less probability, in the third level ($l_3$), while the highest congestion level is never predicted. Similar trends are observed for the other levels. Interestingly, in the case of the highest congestion, $l_4$, the correct detection probability is much higher, reaching 0.8.

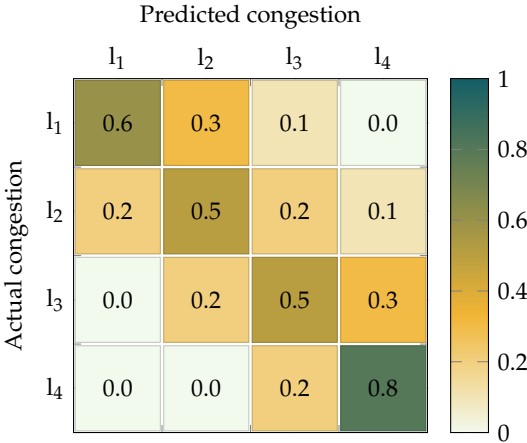

**Figure 10.** Confusion matrix of the hierarchical clustering for a scenario with data rate of 30 `Mbps` and distance of 40 m.

Then, Figure 11 depicts the pseudo-precision observed for the different algorithms. We show the performance obtained for all the scenarios, but those with short distances between user and base

station, for which the performance is very poor, due to the extremely low correlation that was discussed in previous sections. We note that in the case of the RNN algorithm, results could not be obtained for data rates higher than 200 `Mbps`, since the algorithm did not converge. The analysis of this behavior and other neural-network models will be tackled in our future work.

As a general comment, we can observe that for the scenarios that we consider, the distance does not have a strong impact over the pseudo-precision metric. On the other hand, the results also yield a clear relationship between such precision and the data rate. In fact, as the application data rate increases, the precision clearly decreases.

In addition, we can observe that the hierarchical *k-means* performance is higher than that obtained with the HMM and RNN solutions. In turn, the algorithm based on Hidden Markov Chains slightly outperforms that based on neural networks. Finally, as was expected, the supervised solution outperforms the other options, and shows a much more stable behavior as we increase the data rate.

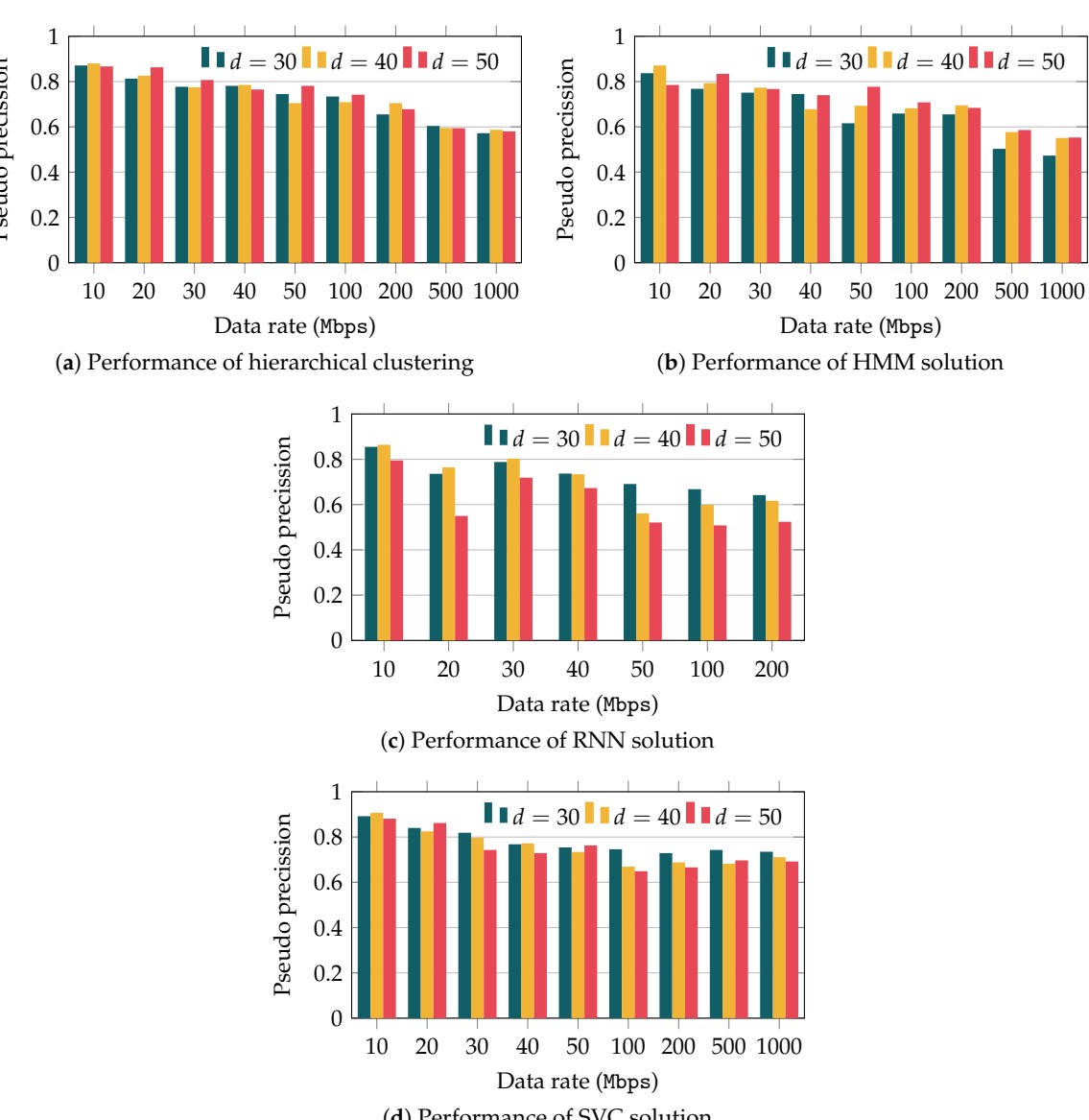

**Figure 11.** Comparison of the different algorithm for multi-level congestion.

## 7. Conclusions

The performance of transport protocols over wireless channels might be severely hindered by punctual congestion situations. These will become more relevant with the consolidation of 5G networks

and, in particular, mmWave links. The scientific community is thus seeking new solutions that can appropriately deal with these situations. In this work, we have studied the potential benefits that Machine Learning techniques could bring. Using realistic channel traces, we have first identified metrics that could be used to predict the congestion level, while being observable from the transport layer. We have studied the correlation of both the delay and the inter-arrival time with the congestion level, which is established by looking at the RLC buffer occupancy.

Afterwards, we assessed whether unsupervised and supervised ML techniques can appropriately predict congestion status. The latter are used as a benchmark, since the objective would be using unsupervised solutions, which impose less requirements and challenges towards an eventual implementation. The results show that the techniques that we analyzed yield reasonable precision, and they could thus be exploited in the design and implementation of congestion-control techniques suited for 5G networks. We also saw that the use of multi-level schemes does not bring a clear benefit, so enforcing a high granularity when identifying channel congestion levels might not be sensible.

We saw that the performance of *k-means* to perform binary classification is rather good, comparable to the supervised approach (SVC), but for high data rates. Furthermore, in multi-level congestion prediction (we established 4 different levels), the behavior of the hierarchical clustering based on *k-means* is again comparable by that exhibited by supervised approaches (SVC and HMM), while the results of the RNN was relatively worse, and it had some convergence issues.

There are three main lines of work that we will foster in our future research. We will first aim at improving the performance of the ML techniques, by considering different metrics and their combinations. In this regard, we will study non-linear relationship between transport layer metrics and congestion (such as mutual information) to elaborate more suitable metrics. This would be especially relevant for high data rates, where the obtained results exhibit poor linear relationship. In addition, we will explore other ML algorithms, keeping in mind the need to keep the complexity low. In particular, we will analyze how to enhance the convergence of the RNN algorithm, for instance using Gated Recurrent Units (GRU) mechanism, instead of LSTM. Furthermore, we will also broaden the scenario, by including heterogeneous deployments (LTE and 5G) leveraging MPTCP, more users (to better assess the influence of the scheduler, which would need to allocate resources), and we will include congestion situations on the transport part of the network, to challenge the learning capacity of the ML solutions, which would need to identify the congestion cause (access or transport segment).

**Author Contributions:** Conceptualization, L.D., M.K., Y.Z. and R.A.; methodology, L.D., M.K., Y.Z. and R.A.; software, A.F.; validation, L.D., and A.F.; formal analysis, A.F.; writing—original draft preparation, A.F. and L.D.; writing—review and editing, L.D., M.K. and Y.Z.; funding acquisition. All authors have read and agreed to the published version of the manuscript.

**Funding:** This research was funded by Ministerio de Economía, Industria y Competitividad (MINECO), Gobierno de España, grant number RTI2018-093475-A-I00.

**Acknowledgments:** This work was supported by the Spanish Government (MINECO) by means of the project FIERCE "Future Internet Enabled Resilient smart CitiEs" under Grant Agreement No. RTI2018-093475-A-I00.

**Conflicts of Interest:** The authors declare no conflict of interest.

## Abbreviations

The following abbreviations are used in this manuscript:

| | |
|---|---|
| ML | Machine Learning |
| mmWave | Millimeter Wave |
| TCP | Transmission Control Protocol |
| PCC | Performance-oriented Congestion Control |
| BBR | Bottleneck Bandwidth and Round-trip propagation time |
| RTT | Round-Trip Time |
| AQM | Active Queue Management |
| MPTCP | Multi-path TCP |

| OBR | Offloading By Restriction |
|---|---|
| QUIC | Quick UDP Internet Connections |
| RLC | Radio Link Control |
| MCS | Modulation and Coding Scheme |
| SDU | Service Data Unit |
| PDCP | Packet Data Convergence Protocol |
| ARQ | Automatic Repeat Request |
| HARQ | Hybrid ARQ |
| RTO | Retransmission Timeout |
| UDP | User Datagram Protocol |
| IaT | Inter-arrival Time |
| OFDMA | Orthogonal Frequency-Division Multiple Access |
| RTT | Round-Trip Time |
| SMA | Simple Moving Average |
| WCSS | Within-cluster Sum of Squares |
| HMM | Hidden Markov Model |
| SVM | Support Vector Machine |
| SVC | Support Vector Classification |
| RNN | Recurrent Neural Networks |
| EM | Expectation-Maximization |
| LSTM | Long Short-Term Memory |
| MSE | Mean Square Error |
| GRU | Gated recurrent units |

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
