# Peer review of "Can We Exploit Machine Learning to Predict Congestion over mmWave 5G Channels?"

_applsci, doi:10.3390/app10186164_

Round 1

Reviewer 1 Report

The authors examine the feasibility of using machine learning techniques to predict the congestion status of 5G access networks. The paper is an interest read, is generally well-written, and studies a timely and important topic. Importantly, the paper is written in a way that it targets both specialists and generalists. The reviewer has only a few comments to be addressed:

  • While the authors have used UDP in their data generation process, the reviewer wonders how the results would be altered if TCP was used. Please discuss.
  • The authors have used the Pearson’s correlation coefficient, which allows to identify linear correlation among the given datasets. Please discuss why it is not necessary to consider the nonlinear correlation.
  • The objectives of the paper are not clearly stated in the introduction section. Please rework.
  • Figure 7 is key to select the metrics that are cross-correlated the most. The reviewer suggests that the authors provide a more in-depth interpretation of Fig. 7. 
  • Given that there has been a growing interest from academia in developing machine learning-based techniques for 5G networks, can the authors include more recently published related work in Section II?
  • The authors have selected a number of supervised and unsupervised machine learning methods, though this selection is not properly justified. It is not clear why the authors have picked , for instance, K-means algorithm among the many other algorithms out there. Please discuss.
  • Even though the authors have provided the list of abbreviations at the end of the manuscript, the reviewer suggests that the authors expand the abbreviations for the first time they appear in the text, e.g., ARQ, FEC.

Author Response

Dear referee, we appreciate you timely review.

In the new version of the manuscript we have tackled all your comments and in the attached document we provide detailed answers. Your comments are the noes referred to "REVIEWER 1". We hope that you find the changes satisfactory.

In the uploaded manuscript document you will find a version with tracked changes, and then the clean document.

Reviewer 2 Report

This work is about identifying features at the transport layer, and deploying machine learning algorithms to predict the congestion level in mmWave 5G channels. The RLC buffer length is considered as the measure of the congestion level, and the congestion is experienced due to mmWave channel variations.

1- Correlation coefficient is a measure of a linear relationship between two variables. That is probably the reason why the delay and the congestion have a low correlation at high data rates?. For example, if the delay and the buffer length have a non-linear relationship such as an exponential relationship then the correlation would not be a good indicator of dependency. I think mutual information would be a better metric as it can capture non-linear dependency, and it can also measure redundancy. It seems to me mutual information is a better metric to analyse the problem.

2- In Figure 4, what is the reason for seeing a different behaviour for the distances of 10 and 20 meters at the data rate of 30 Mbps.

3- I think this work needs to be more clear about the RNN part. It is not clear how LSTM has beed deployed including the input features, and sequence. The LSTM can be used for classification. What is the reason for this sentence "this solution predicts a continuous congestion level, instead of a discrete one."?. There are some properties for the sequences on which LSTM can be successful such not having a trend or a seasonality component (if the sequence is short), and being stationary. I think GRU would be a better option for shorter sequences.

4- Typo in defining variance on page 9.

5- Wondering if we can formulate the problem as a forecasting problem, i.e., using the behaviour that we see up to the time instance n to forecast the congestion level at future time instances.

Author Response

Dear referee, we appreciate you timely review.

In the new version of the manuscript we have tackled all your comments and in the attached document we provide detailed answers. Your comments are the noes referred to "REVIEWER 2". We hope that you find the changes satisfactory.

In the uploaded manuscript document you will find a version with tracked changes, and then the clean document.

Round 2

Reviewer 2 Report

Although some of my comments have been considered as future works, I would recommend the paper to be accepted.